# Public investments in the clinical development of bedaquiline

**Dzintars Gotham**[1]*, **Lindsay McKenna**[2], **Mike Frick**[2], **Erica Lessem**[2]

**1** Independent Researcher, London, United Kingdom, **2** Treatment Action Group, New York, New York, United States of America

* dzintarsgotham@gmail.com

## Abstract

### Introduction

In 2012, bedaquiline became the first new treatment from a novel class to be approved for tuberculosis in nearly five decades and is now a core component of the standard of care for multidrug-resistant tuberculosis. In addition to the originator pharmaceutical company, Janssen, a range of governmental and non-profit entities have contributed to the development of bedaquiline.

### Materials and methods

We identified various avenues of public investments in the development of bedaquiline: direct funding of clinical trials and a donation programme, tax credits and deductions, and revenues resulting from the priority review voucher (PRV) awarded to the originator. Data on investments were gathered through contact with study leads and/or funders; for non-responses, published average costs were substituted. The originator company's expenses were estimated by similar methods. Tax credits and deductions were calculated based on estimated originator trial costs and donation expenses. The value of the PRV was estimated by application of a published model.

### Results

Public contributions through clinical trials funding were estimated at US$109–252 million, tax credits at US$22–36 million, tax deductions at US$8–27 million, administration of a donation programme at US$5 million, PRV revenues at US$300–400 million. Total public investments were US$455–747 million and originator investments were US$90–240 million (if capitalized and risk-adjusted, US$647–1,201 million and US$292–772 million, respectively).

### Conclusions

Estimating the investments in the development of a medicine can inform discussions regarding fair pricing and future drug development. We estimated that total public investments exceeded the originator's by a factor of 1.6–5.1.

**Data Availability Statement:** Data are available in the paper and Supporting Information files.

**Funding:** This analysis was funded by Treatment Action Group (TAG), New York (http://www.treatmentactiongroup.org/). There are no

associated grant numbers. The funder provided support in the form of salaries for authors LM, MF, EL, and a grant to DG working as an independent researcher, but did not have any additional role in the study design, data collection and analysis, decision to publish, or preparation of the manuscript. The specific roles of these authors are articulated in the 'author contributions' section.

**Competing interests:** LM, EL, and MF are employees of Treatment Action Group. DG has received a research grants from Treatment Action Group for this analysis. The authors declare no other potential conflicts of interest.

## Introduction

Tuberculosis (TB) caused 1.5 million deaths in 2018, making TB the world's leading infectious cause of death, ahead of HIV/AIDS. There were 484,000 incident cases of multidrug-resistant TB (MDR-TB) in 2018, of which only 32% were started on treatment. Among those who received treatment, only 56% of people with drug-resistant TB and 39% of people with extensively drug-resistant (XDR) experienced a successful treatment outcome [1].

The approval of bedaquiline by the US Food and Drug Administration (FDA) in 2012 made it the first new treatment for TB approved in nearly five decades [2]. Bedaquiline, a daily oral medicine, represents a significant improvement for the treatment of MDR-TB. Historically, MDR-TB has been significantly harder to treat than drug-sensitive TB: For decades, treatment of MDR-TB required long (often 20 months or more) treatment with a combination of medicines, including daily injections, which carry a high risk of severe adverse effects such as hearing loss. In early 2018, the WHO published new treatment guidelines establishing a new all-oral, bedaquiline-based standard of care for MDR-TB [3]. These newer bedaquiline-containing regimens are all-oral, improve DR-TB cure rates, and allow for the replacement of medicines that cause severe adverse effects [4, 5].

The bedaquiline compound was identified in 2005 by Janssen, a subsidiary of the pharmaceutical company Johnson & Johnson [6]. Apart from the originator company, numerous other actors in academia, non-governmental and humanitarian organisations, and governments have played and continue to play key roles in the development of bedaquiline. In 2009, Janssen entered into a collaboration with the Global Alliance for TB Drug Development (TB Alliance), a non-profit organisation, to share resources and expertise in further development [7]. After the accelerated approval of bedaquiline in the US (2012) and EU (2013), both of which were based on Phase 2 originator trials, a range of clinical questions remained that would need to be addressed through research before bedaquiline could be used to its full potential. Several Phase 3 trials and real-world cohort studies to confirm the efficacy of bedaquiline and to identify optimal combinations with other TB medicines are being undertaken by non-profit actors including TB Alliance, the Union (previously the International Union Against Tuberculosis and Lung Disease), Médecins Sans Frontières (MSF), and Partners In Health (PIH).

Apart from clinical trial costs, other indirect public contributions have supported the development of bedaquiline. These include, potentially, tax breaks and regulatory incentives awarded to the originator.

The aim of this analysis was to quantify the expenditures and incentives put towards the development of bedaquiline by the public sector. Quantifying these investments can contribute to debates concerning the pricing of bedaquiline, the role of the public sector in pharmaceutical research and development (R&D), and the costs of bringing a novel medicine to market.

## Methods

To quantify public sector contributions to bedaquiline R&D and roll-out, we identified various avenues of public investments in the development of bedaquiline: direct funding of clinical trials and a donation programme, tax credits and deductions, and revenues resulting from the priority review voucher (PRV) awarded to the originator. A variety of methodological approaches were used, which are outlined below. Inflation adjustments were made to 2018 US dollars using the US gross domestic product deflator reported by the World Bank [8]. Currencies were converted to US dollars using the historical annual exchange rates (S1 File).

## Clinical trials

Relevant clinical trials were identified by searching for trials including bedaquiline as an intervention in the ClinicalTrials.gov database [9]. Data on trial costs were requested by email from study leads or representatives (S1 File). Where exact figures were reported to be unavailable, best estimates were requested.

Where information on trial costs could not be obtained in this way, we used reported phase-specific average clinical trial costs for anti-infectious drug candidates in the US (Sertkaya et al.; see also S1 File) [10]. Notably, requests to TB Alliance and Janssen for information on trial expenses were not fulfilled by the time of journal submission. We generated a range of clinical trial cost estimates by using these estimates as the maximum of the range, and calculating a lower-bound estimate that would account for lower clinical trial costs in LMICs compared to the US, assumed to be 40% lower based on reported differences to costs in China and India [11], and the proportion of costs attributable to bedaquiline development.

Some of the included trials have additional clinical research questions not limited to bedaquiline—for example, some trials are also assessing delamanid, another new MDR-TB drug. The extent to which the costs of such trials are counted in the overall expenditures on bedaquiline development can be viewed in two different ways. On the one hand, we could ascribe only a proportion of the overall cost to bedaquiline where the trial has other key investigational foci. On the other hand, as bedaquiline is not yet fully integrated into TB treatment programmes, all trials involving bedaquiline are arguably key to its clinical development, especially in the context of TB research being increasingly regimen-focussed rather than drug-focussed. We thus considered that a range was appropriate, where the maximum is unadjusted for proportional attribution to bedaquiline, while the minimum is adjusted based on the authors' judgement (S1 File).

## Comparison to originator's R&D expenditures

A representative of Janssen has stated that the company's R&D expenditures for bedaquiline were approximately US$500 million [12]. However, the company has not provided a breakdown of this number. In order to examine this figure, and compare it to public expenditures, we estimated Janssen's R&D expenditures using the methods for clinical trial cost estimates described above, additionally calculating risk-adjusted costs and capitalized costs, as industry R&D cost estimates often include these elements (S1 File). Risk-adjusting costs is intended to capture the lost investments in alternative drug candidates that failed clinical trials, and capitalization of costs is intended to capture the 'opportunity cost' of not investing the money spent on drug development differently, for example, in an index fund.

The term 'out-of-pocket' designates (estimates of) actual expenditures, that is, expenditures that are not risk-adjusted or capitalized.

## Orphan drug incentives

Legislation in the US, EU, and Japan has created financial incentives for the development of medicines for rare diseases, so-called orphan drugs. Orphan drug designation confers a number of monetary benefits to the relevant sponsor company, including orphan drug tax credits (ODTC), waiving of regulatory fees, accelerated approval, and added market exclusivity. As these financial incentives offset the originator's R&D expenditures, we included these in our survey. We considered the ODTC to be the main relevant public investment and estimated its value (S1 File).

While the accelerated approval granted to bedaquiline by the FDA likely translates to additional revenues, this represents a regulatory body (in this case) correctly prioritizing among

submitted applications for public health purposes and therefore did not consider this to be a public contribution to R&D costs.

We considered that market exclusivity associated with orphan drug designation is not expected to increase lifetime sales and that waivers of FDA and EMA fees likely has a net negligible effect on the total public contributions to bedaquiline development (S1 File).

### Donation programme

Key real-world evidence that led to bedaquiline becoming the backbone of the preferred treatment for MDR-TB in WHO guidelines relied in part on donated bedaquiline. We therefore considered public investments in the donation programme to be investments in bedaquiline development.

Donations of pharmaceutical products can be tax deductible in the US [13, 14], and we assumed that Janssen applied these deductions. Over 2015–2019, Janssen has offered donations of 105,000 treatment courses of bedaquiline. We assumed that the tax-deductible amount per donated treatment course was US$266 (S1 File).

### Priority review voucher

Priority review vouchers (PRVs) are a regulatory reward mechanism established in the US in 2007 with the aim of encouraging the development of drugs for neglected diseases [15]. The FDA grants PRVs to sponsors of successful new drug applications (NDA) for treatments for neglected diseases that meet certain criteria. The PRV can then be used by the holder, at a later date, to secure priority (i.e. faster than normal) review of any other NDA. The holder can also sell PRVs to another company, which can then use it to secure priority review for any of its NDAs. Due to the commercial benefits of securing priority review from the FDA, PRVs have been sold for US$68–350 million [15].

We consider the award of a PRV to be a public investment in R&D. The value of the PRV to the holder is not directly paid to the drug developer out of the public purse. However, the monetary value of the PRV derives from the increased period of sales for the product that is approved more quickly through use of the PRV, that is, through additional expenditures on the 'prioritized' drug by the US public.

Janssen was awarded a PRV for the approval of bedaquiline, and later applied this PRV in 2017 to expedite regulatory review of guselkumab, a treatment Janssen developed for plaque psoriasis [16]. We applied a simplified version of a published model to estimate the commercial value of the PRV (S1 File) [15].

### Technical assistance and cohort studies

We explored the possibility of quantifying public investments in technical assistance work that has supported bedaquiline roll-out (e.g. national registration, guideline implementation) by reviewing publicly available data and discussing the feasibility with organizations known to be active in technical assistance work for MDR-TB, including USAID and Médecins Sans Frontières (MSF). We ultimately considered it not feasible at present to attribute a certain monetary value to technical assistance work on bedaquiline roll-out.

Similarly, we explored the possibility of including key cohort studies, which have assessed the use of bedaquiline in real-world settings (generally within compassionate use programmes), by contacting study leads. However, patients in these cohorts received bedaquiline as part of normal healthcare operations (e.g. in MSF projects), and after discussion with study leads, we concluded that identifying the costs attributable to bedaquiline research alone would not be feasible.

### Role of the funding source

This study was funded from the operating budget of Treatment Action Group (TAG), a non-profit research and policy think tank focussed on HIV, tuberculosis, and hepatitis C. Three of the authors of this study are employees of TAG. The corresponding author had full access to all the data in the study and had final responsibility for the decision to submit for publication.

## Results

Table 1 and Fig 1 summarise monetary public contributions to the clinical development of bedaquiline. Taking all the identified investments together, we estimated that total public expenditures have been 3.1–5.1 times those of the originator (US$455–747 million versus US$90–240 million), or 1.6–2.2 (US$647–1,201 million versus US$292–772 million) when the cost of failures and costs of forgoing other investment opportunities are counted (Table 1 and S1 File).

Table 2 shows clinical trials on bedaquiline that received direct public funding. The total amount of public funding for bedaquiline clinical trials was US$109–252 million (out-of-pocket, that is, not risk-adjusted or capitalized). Estimated public contributions were US$7.0–10.9 million for Phase 1, US$36.3–90.8 million for Phase 2, US$60.8–146.5 million for Phase 3, and US$15.5–31.0 million for Phase 4 trials (where a trial spans two phase categories, expenditure attributed equally across categories). Estimated public and originator expenditures on clinical trials, by year, are shown in Fig 2.

We estimated that the potential rewards to Janssen in the form of orphan drug tax credits ranged $22–36 million, deriving from clinical trials undertaken between 2005 and 2012 (S1 File).

Originator expenses for the donation programme were estimated at US$14–77 million, and reductions in tax bill resulting from tax deductions were estimated at US$8–27 million (S1 File). The US Government has contributed US$5 million to the donation programme [17].

We estimated that the PRV translates to US$300–400 million in additional revenue for Janssen (S1 File).

**Table 1. Comparison of overall estimated public and originator investments (2018 US$ millions).**

|  | Public | Originator | Ratio of public to originator expenditures* |
|---|---|---|---|
| Clinical trials |  |  |  |
| *Out of pocket* | 120–279 | 76–163 | 1.6–1.7 |
| *Capitalized* | 142–328 | 115–280 | 1.17–1.23 |
| *Capitalized and risk-adjusted* | 312–733 | 278–695 | 1.05–1.12 |
| Funding through PRV | 300–400 |  |  |
| Orphan drug tax credit | 22–36 |  |  |
| Bedaquiline donation program | 13–32† | 14–77 | 0.4–0.9 |
| **Totals** |  |  |  |
| **Out-of-pocket expenditures** | **455–747** | **90–240** | **3.1–5.1** |
| **Capitalized and risk-adjusted expenditures** | **647–1,201** | **292–772** | **1.6–2.2** |

PRV—priority review voucher.

*Ranges for ratios are calculated as the bottom of the range for public funding divided by bottom of the range for Janssen funding, and top of the range for public funding divided by top of the range for originator funding.

†Composed of US$8–27 million through tax deductions for originator and US$5 million through public funding of administration of the donation programme.

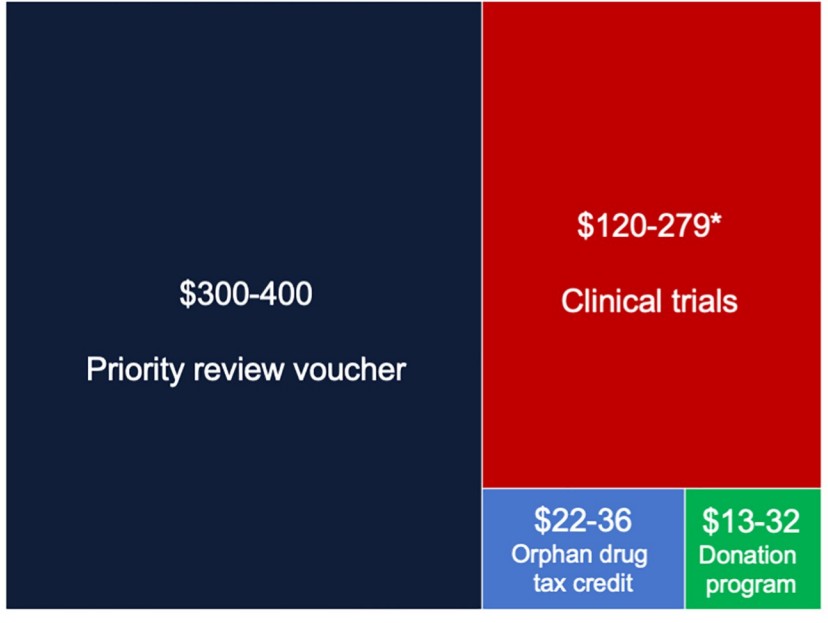
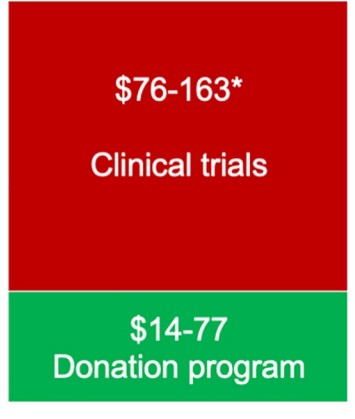

**Fig 1. Public sector investments and incentives for the development and roll-out of bedaquiline.** Segments scaled by area. Mid-point of ranges used for area.

Our estimates of originator expenditures are substantially lower than the originator's stated costs of 'approximately US$500 million', except when costs are risk-adjusted and capitalized, in which case the US$500 million figure falls within our range of estimates (Table 1). Estimated public sector clinical trial expenditures were 1.6–1.7 times the originator's (out-of-pocket expenditures). When compared 'apples to apples'–applying capitalization and risk-adjustment to in the same manner to both originator and public expenditures—public sector and originator clinical trial expenditures were similar. This is mainly due to the fact that originator trials took place years before most public sector trials, and capitalization exponentially augments expenses that are further in the past.

## Discussion

We analysed investments in the development of bedaquiline, the first new TB treatment to be approved in 50 years [2]. Overall public investments were estimated at US$451–742 million, comprising direct clinical trial funding of US$120–279 million, tax credits at US$22–36 million, tax deductions at US$8–27 million, costs of donation programme administration at US$5 million, and future revenues from the PRV at US$300–400 million. Clinical trial expenditures by the public sector were 60–70% higher compared to the originator's when out-of-pocket costs are compared, but 5–12% higher when figures are capitalized and risk-adjusted. However, when additional investments—PRV, tax credits, and donation program costs—are included, we estimate that total public expenditures have been 3.1–5.1 times those of the

**Table 2. Public expenditures on bedaquiline clinical trials.**

| Trial phase | Short title | Description | Sponsor(s) | Patients enrolled | Study start and end dates | Trial cost (2018 US$ million)[A] |
|---|---|---|---|---|---|---|
| 1 | ACTG 5267[B] | Interaction of bedaquiline and efavirenz | NIAID | 37 | 2009–10 | 0.4 |
| 1 | TMC207-CL002 | Interaction of bedaquiline and rifapentine or rifampicin | TB Alliance | 32 | 2010–10 | 2.9–4.9* |
| 1 | TMC207 +/- Rifabutin/ Rifampin | Interaction of bedaquiline and rifabutin or rifampicin | NIAID | 33 | 2011–12 | 2.9–4.9* |
| 1 | TASK-002 | Bioequivalence of crushed bedaquiline tablet | IMPAACT, NIAID, NICHD, NIMH | 24 | 2016–17 | 0.2 |
| 1/2 | IMPAACT 1108 | PK, safety, tolerability of bedaquiline in infants, children, adolescents | NIAID | 72 | 2017–22 | 1.0[C] |
| 2 | TMC207-CL001 | Early bactericidal activity of bedaquiline | TB Alliance | 68 | 2010–10 | 9.9–16.5* |
| 2 | NC-001 | Early bactericidal activity of bedaquiline with pretomanid, moxifloxacin, and pyrazinamide | TB Alliance | 85 | 2010–11 | 5.0–16.5* |
| 2 | NC-003 | Early bactericidal activity of bedaquiline with pretomanid, clofazimine, and pyrazinamide | TB Alliance | 105 | 2012–13 | 5.0–16.5* |
| 2 | NC-005 | Early bactericidal activity of bedaquiline with pretomanid, moxifloxacin, and pyrazinamide | TB Alliance | 240 | 2014–18 | 5.0–16.5* |
| 2 | ACTG 5343 | PK, safety, tolerability of bedaquiline and delamanid alone and in combination | NIAID | 84 | 2016–20 | 1.1–2.2 |
| 2 | Janssen C211 | PK of bedaquiline in children and adolescents | Janssen, Unitaid | 60 | 2016–25 | 1.5 |
| 2 | IMPAACT P1108 | PK of bedaquiline in children and adolescents | NIAID, NICHD | 72 | 2017–22 | 1.0† |
| 2 | SimpliciTB (B-Pa-M-Z) NC-008 | Efficacy, safety, tolerability of bedaquiline with pretomanid, moxifloxacin, and pyrazinamide for DS- and DR-TB | TB Alliance | 450 | 2018–22 | 6.5–21.6* |
| 2/3 | NEXT | Open-label study of a bedaquiline-containing regimen for DR-TB | UCT, UoL, WSU, UoS, UCTLI | 300 | 2015–19 | 3.8† |
| 2/3 | TB-PRACTECAL | Study of regimens containing bedaquiline for DR-TB | MSF, TB Alliance, DNDi, others | 630 | 2017–21 | 4.0–8.0 |
| 2/3 | TRUNCATE-TB | Study of regimens containing bedaquiline for DS-TB | UCL, NUHS, SCRI | 900 | 2018–22 | 1.5–7.4† |
| 3 | NiX-TB | Study of bedaquiline with pretomanid and linezolid for DR-TB | TB Alliance | 109 | 2015–21 | 8.0–26.6* |
| 3 | STREAM Stage 2 | Study of regimens containing bedaquiline for DR-TB | The Union, UK MRC | 1155 | 2016–21 | 20.0–40.0 |
| 3 | endTB interventional | Study of regimens containing bedaquiline and/or delamanid for DR-TB | MSF, PIH, others | 750 | 2016–21 | 10.0–19.9 |
| 3 | ZeNix (B-Pa-L) NC-007 | Study of bedaquiline with pretomanid and linezolid in XDR-TB and treatment intolerant or non-responsive MDR-TB | TB Alliance | 180 | 2017–22 | 8.0–26.6* |
| 3 | endTB-Q | Study of regimens containing bedaquiline for fluoroquinolone-resistant MDR-TB | MSF, PIH, others | 500 | 2019–22 | 6.6–13.1 |
| 4 | endTB observational | Observational study of real-world use of bedaquiline- and delamanid-containing regimens for DR-TB | MSF, PIH, others | 2600 | 2016–20 | 15.5–31.0 |

DR-TB—drug-resistant TB. DS-TB—drug-sensitive TB. PK—pharmacokinetics. XDR-TB—extensively drug-resistant TB.

CAPRISA—Centre for the AIDS Programme of Research in South Africa.

IMPAACT—International Maternal Pediatric Adolescent AIDS Clinical Trials Group.

NICHD—US National Institute of Child Health and Human Development.

NIAID—National Institute of Allergy and Infectious Diseases.

UCT—University of Cape Town.

UoL—University of Limpopo.

WSU—Walter Sisulu University.

UoS—University of Stellenbosch.

UCTLI—University of Cape Town Lung Institute.

Phase, enrolment, and date data from the ClinicalTrials.gov database.

[A] Where reported as a range, the lowest end of the range represents trial costs for the bedaquiline-attributable portion. Sources for trial cost data are given in the Appendix (S1 File).

[B] This study likely satisfied the requirement placed on Janssen by the FDA for a trial studying interaction with efavirenz (see appendix).

[C] The costs of the trial were cited as "just under 1 million".

[D] Contributed by Unitaid via the TB Alliance STEP-TB project.

* Study costs estimated based on Sertkaya et al., with the lower end of the range representing a 40% reduction to account for lower trial costs in LMICs.

† The given values are best-guess estimates by investigators leading the trial.

Trials run by pharmaceutical companies are not shown, other than C211.

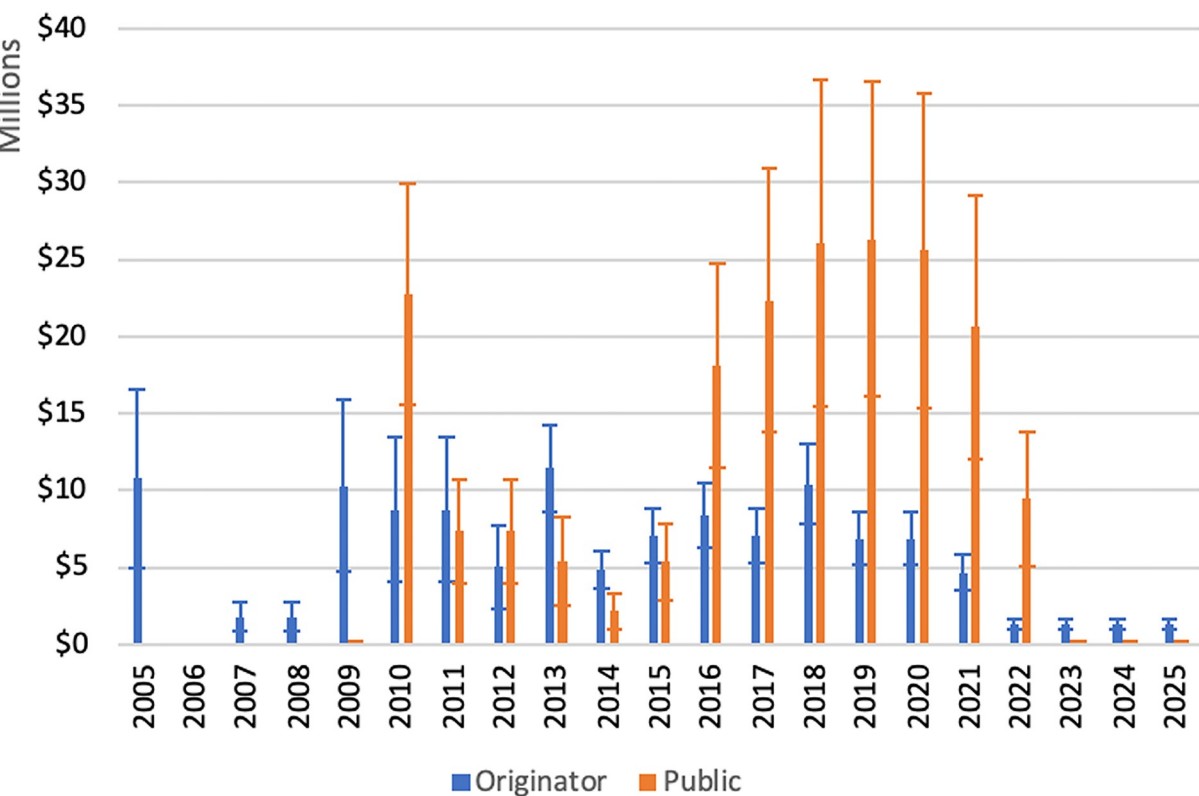

**Fig 2. Estimated public and private expenditures on bedaquiline clinical trials, by year.** Error bars represent the range of estimates produced as described in Methods. Values shown are for out-of-pocket expenditures. Year-by-year comparisons of out-of-pocket, risk-adjusted and capitalized costs are available in the Appendix (S1 File).

originator, or 1.6–2.2 when the cost of failures and costs of forgoing other investment opportunities are counted (S1 File).

The larger questions posed by these findings relate to determining the desirable or fair balance between the R&D investments of the private sector, the R&D investments of the public sector, and the prices paid by health systems and patients. In more practical terms: where in the drug development process for a neglected disease medicine the public should 'step in', and how should these public investments be reflected in the prices of medicines?

Bedaquiline represents a success story for the feasibility and effectiveness of clinical development done through publicly funded projects. Between 2005 and 2012, the originator undertook five Phase 1 trials and three Phase 2 trials, securing FDA and EMA "accelerated" approvals, meaning the regulators granted approval with the requirement that further confirmatory trials be undertaken. At the time of approval, bedaquiline was seen as a promising but by no means established treatment—a range of research gaps remained to be addressed, in addition to concerns about safety. Following the accelerated approvals, essentially all of the clinical development has occurred in publicly funded projects. Late 2012 or early 2013 can thus be seen as a point of 'handing off' development from the originator to the public sector. (The public sector was also involved prior to this, with three Phase 1 and two Phase 2 public sector trials run starting before 2012.).

In general, little information is available on pharmaceutical R&D expenditures—both in the public and private sectors—as private entities do not publish details on R&D expenditures, and public sector entities, with some exceptions, do not analyse research grant data to attribute

expenditures to resulting health products. Available estimates of R&D expenditure range widely, and some have been criticised for being non-transparent in their methodologies and using potentially biased samples [18, 19].

While a representative of Janssen has noted bedaquiline R&D expenditures of US$500 million [12], no details are available on the breakdown of this figure. Based on our estimates, we conclude that this figure is likely a risk-adjusted and capitalized cost estimate, while we estimated Janssen's direct (out of pocket) expenditures on clinical trials to be substantially lower, at US$76–163 million. Our estimates suggest that across the (future) value of the PRV and orphan drug tax credits, Janssen's R&D investments will likely be fully compensated or overcompensated.

This analysis can be seen as a case study within the broader discussion of drug development in neglected disease areas. For diseases that predominantly affect the global poor, such as TB, there is a far lesser commercial incentive to invest in R&D compared to diseases that also affect populations in high-income countries, such as lung cancer and cardiovascular disease. This has led to dramatic imbalances between R&D investments and disease burden, with so-called neglected diseases receiving about 10 times less R&D investment than would be expected based on the attributable global disease burden [20]. The costs estimated in this analysis can inform further initiatives to fund drug development for neglected diseases. Drugs for Neglected Diseases initiative (DNDi), the largest non-profit drug development initiative for neglected diseases, has estimated that the cost of developing a novel medicine (new chemical entity) to approval is EUR 44–71 million out-of-pocket, or EUR 60–190 million including the costs of failures [21]. These numbers are about significantly lower than our estimates of total (public and private) expenditures on bedaquiline clinical trials. This difference may be explained by our inclusion of post-approval trials, the large number of trials needed for bedaquiline to describe its role as part of various regimens and in various settings, and long follow-up times needed.

Despite the significant improvement in MDR-TB treatment outcomes offered by bedaquiline, by September 2019, only 36,000 treatment courses of bedaquiline had been received by patients worldwide, compared to the estimated 160,000 patients needing bedaquiline annually [22]. Various factors have limited patient access to bedaquiline, including the limited data and clinical experience available at the time of approval by regulators in the US and EU, the prices charged in countries not eligible for the donation programme, the limited scope of use recommended in initial guidelines, and a lag in registration with national medicines regulatory agencies.

The USAID/Janssen donation programme for bedaquiline was first announced in December 2014, two years after approval in the US (prior to this, some bedaquiline was donated by Janssen through compassionate use programmes). Outside of the donation programme, Janssen has used a tiered pricing approach, setting the price of six-months of bedaquiline at US$30,000 per treatment course in high-income countries, US$3,000 in upper-middle-income countries, and US$900 for 'least developed/resource-limited countries' [23], more recently announcing an agreement to provide bedaquiline for US$400 per six-month treatment course for low- and most middle-income countries eligible to procure TB medicines from the Global Drug Facility [24]. At the same time, it was independently estimated that the cost of production for bedaquiline is around US$130 per treatment course [25], and a representative of Janssen has noted that approximately one-third of the US$400 price (~US$133) covers the cost of manufacture and distribution (the other two-thirds covering regulatory and pharmacovigilance costs, as well as 'programmatic' expenses, such as for ensuring appropriate use) [12]. Janssen's donation programme for bedaquiline ended in March 2019, and activists have called

for bedaquiline to be priced at $180 per treatment course, as well as for transparency in R&D investments [26].

## Limitations

Pre-clinical investments were not assessed. These may include investments in laboratory-based basic research and animal studies to identify bedaquiline as a promising lead compound for testing in humans, and would arguably include both investments by the originator and investments by the public, such as work done by the US Army, to identify the potential efficacy of the broader family of compounds in treating tuberculosis [27].

Where trial costs were reported to us, we asked for overall trial costs and did not attempt a detailed accounting exercise. Our estimates also relied, in part, on average clinical trial costs reported by a US-based industry analysis group [10]. While these average costs were Phase-specific and were adjusted for potentially lower trial costs in LMICs and proportion attributable to bedaquiline, costs were not adjusted to take into account different trial characteristics such as enrolment numbers or duration.

We did not attempt to quantify the investments that governments in countries with high TB burdens may be making into bedaquiline roll-out domestically, instead focusing on large, centralised initiatives. Public investments in technical assistance work and cohort studies were not captured.

## Conclusions

Substantial public investments have been made in the development of bedaquiline, estimated at US$455–747 million, of which direct clinical trial funding made up an estimated US$120–279 million. We estimated that public sector expenditures have exceeded expenditures by the originator by a factor of 3.1–5.1, or 1.6–2.2 when the cost of failures and costs of forgoing other investment opportunities are counted.

## Supporting information

**S1 File. Appendix.**
(PDF)

**S2 File. Clinical trial cost estimate dataset.**
(XLSX)

## Acknowledgments

We thank the experts who provided thoughtful comments on drafts of this analysis: Jennifer Reid (Médecins Sans Frontières), Katy Athersuch (Médecins Sans Frontières), Suerie Moon (Graduate Institute Geneva), Manuel Martin (Médecins Sans Frontières), and Nicholas Lusiani (Oxfam America). We are grateful to the respondents who provided expenditure data (details on personal communications are available in the Appendix, S1 File).

## Author Contributions

**Conceptualization:** Dzintars Gotham, Lindsay McKenna, Mike Frick, Erica Lessem.

**Data curation:** Dzintars Gotham, Lindsay McKenna, Mike Frick, Erica Lessem.

**Funding acquisition:** Lindsay McKenna, Mike Frick, Erica Lessem.

**Investigation:** Dzintars Gotham.

**Methodology:** Dzintars Gotham, Lindsay McKenna, Mike Frick, Erica Lessem.

**Supervision:** Lindsay McKenna, Mike Frick, Erica Lessem.

**Writing – original draft:** Dzintars Gotham.

**Writing – review & editing:** Dzintars Gotham, Lindsay McKenna, Mike Frick, Erica Lessem.

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
