## [Decision Letter · Decision Letter 0]

14 May 2020

PONE-D-19-31233

Public investments in the clinical development of bedaquiline

PLOS ONE

Dear Dr Gotham,

Thank you for submitting your manuscript to PLOS ONE. I do apologise that it took me so long to get your manuscript peer reviewed. After careful consideration, we feel that it has merit but does not fully meet PLOS ONE’s publication criteria as it currently stands. Therefore, we invite you to submit a revised version of the manuscript that addresses the points raised during the review process. We would appreciate receiving your revised manuscript by 31 May 2020.

We look forward to receiving your revised manuscript.

Kind regards,

Marian Loveday, Ph.D.

Academic Editor

PLOS ONE

Journal Requirements:

"This analysis was funded by Treatment Action Group, New York (http://www.treatmentactiongroup.org/). There are no associated grant numbers. Three of the authors of this analysis are employees of TAG."

We note that one or more of the authors are employed by a commercial company: Independent, London, UK.

Reviewers' comments:

Reviewer's Responses to Questions

**Comments to the Author**

1. Is the manuscript technically sound, and do the data support the conclusions?

Reviewer #1: Yes

Reviewer #2: Partly

2. Has the statistical analysis been performed appropriately and rigorously? 

Reviewer #1: N/A

Reviewer #2: Yes

3. Have the authors made all data underlying the findings in their manuscript fully available?

Reviewer #1: Yes

Reviewer #2: Yes

4. Is the manuscript presented in an intelligible fashion and written in standard English?

Reviewer #1: Yes

Reviewer #2: Yes

5. Review Comments to the Author

Reviewer #1: The authors have clearly outlined the sources of R&D funding information, their approach to estimating data where requests for input were ignored, and the components of the R&D funding that have been included and excluded. The results are plainly and clearly presented. The limitations are also clearly identified and their potential scale and direction described.

My only suggestion is that, while the Union is now styled as such, to those not from the TB field, the name may be obscure. The previous full name - International Union Against Tuberculosis and Lung Disease - could be included in parentheses.

Reviewer #2: Congratulations to the authors for putting together such a concise, methodologically detailed and well thought out paper. I thoroughly enjoyed reviewing this paper as it provides a strong and critical argument around the considerations required when setting drug prices. Often R&D is cited as the reason behind the significant gaps between the estimated cost of production, as the and the market value of drugs, and therefore reflecting on public expenditure towards drug development in addition to that of pharmaceutical companies is relevant when negotiating prices for a public good. I wish the authors the best of luck with this paper moving forward.

I just have a few comments and points of clarity:

1. It would be good for some further explanation on how pharmaceutical companies set prices and the justification to set the price for Bedaquilin

2. The authors reference that the cost of production of bedaquilin (lines 403) is $130, whilst Janssen has set the price of Bedaquilin as $400 to to some LMICs, while others pay $900.

3. The authors assume that clinical trial costs in LMICs are assumed to be 40% lower. This is based on a reference comparing the costs in Asia to that of the US and European countries. Can the same be assumed for Africa?

4. The authors reference a speech during the 2018 UN-High Level Meeting on Tuberculosis for the estimate of $500 million on R&D expenditures. Do the authors have any other reference for this estimate or a potentially be able to provide a time for the statement as the video is lengthy.

https://dash.harvard.edu/bitstream/handle/1/37945571/CHAN-THESIS-2018.pdf?sequence=3

5. In the interest of being as objective as possible, it is unlikely that the estimates provided by Janssen would not include costs that are risk adjusted and capitalized and therefore it is more likely that the $500 million estimate by Janssen can be substantiated. Furthermore, I feel that the authors should emphasize that the main finding is that public sector and originator clinical trial expenditures were similar-“comparing apples to apples” as the authors note, would be the most appropriate approach. That still however provides a strong argument for pricing to take into consideration that public sector investment and not allow for a ‘business as usual’ approach by the pharmaceutical company. Should pricing be in place to compensate Janssen for their R&D investments and stimulate further innovation, then the public sector should enjoy the same returns on their investments to stimulate further partnerships with the pharmaceutical industry.

6.Some of the ratios of public to originator expenditures appear incorrect based on the methodology outlined on table 1. I would recheck all of the numbers

e.g

Capitalized clinical costs:

142/115=1.2

328/280=1.2

6. PLOS authors have the option to publish the peer review history of their article (what does this mean?). If published, this will include your full peer review and any attached files.

Reviewer #1: Yes: Andy Gray

Reviewer #2: No

---

## [Author Response · Author response to Decision Letter 0]

21 May 2020

We sincerely thank the reviewers for their time and their thoughtful comments, which have improved the manuscript. Please see attached Word file with responses to review.

---

## [Decision Letter · Decision Letter 1]

1 Sep 2020

Public investments in the clinical development of bedaquiline

PONE-D-19-31233R1

Dear Dr. Gotham,

We’re pleased to inform you that your manuscript has been judged scientifically suitable for publication and will be formally accepted for publication once it meets all outstanding technical requirements.

Kind regards,

Marian Loveday, Ph.D.

Academic Editor

PLOS ONE

---

## [Editor Report · Acceptance letter]

10 Sep 2020

PONE-D-19-31233R1

Public investments in the clinical development of bedaquiline

Dear Dr. Gotham:

I'm pleased to inform you that your manuscript has been deemed suitable for publication in PLOS ONE. Congratulations! Your manuscript is now with our production department.

Kind regards,

on behalf of

Dr. Marian Loveday 

Academic Editor

PLOS ONE